# Outline of Salivary Gland Pathogenesis of Sjögren’s Syndrome and Current Therapeutic Approaches

**DOI:** 10.3390/ijms241311179

**Published:** 2023-07-06

**Authors:** Yoshiaki Yura, Masakazu Hamada

**Affiliations:** Department of Oral & Maxillofacial Oncology and Surgery, Osaka University Graduate School of Dentistry, Osaka 565-0871, Japan; hamada.masakazu.dent@osaka-u.ac.jp

**Keywords:** Sjögren’s syndrome, salivary gland, pathogenesis, immune checkpoint inhibitor, muscarinic agonist, sialendoscopy, biological therapy, monoclonal antibody

## Abstract

Sjögren’s syndrome (SS) is an autoimmune disease characterized by the involvement of exocrine glands such as the salivary and lacrimal glands. The minor salivary glands, from which tissue samples may be obtained, are important for the diagnosis, evaluation of therapeutic efficacy, and genetic analyses of SS. In the onset of SS, autoantigens derived from the salivary glands are recognized by antigen-presenting dendritic cells, leading to the activation of T and B cells, cytokine production, autoantibody production by plasma cells, the formation of ectopic germinal centers, and the destruction of salivary gland epithelial cells. A recent therapeutic approach with immune checkpoint inhibitors for malignant tumors enhances the anti-tumor activity of cytotoxic effector T cells, but also induces SS-like autoimmune disease as an adverse event. In the treatment of xerostomia, muscarinic agonists and salivary gland duct cleansing procedure, as well as sialendoscopy, are expected to ameliorate symptoms. Clinical trials on biological therapy to attenuate the hyperresponsiveness of B cells in SS patients with systemic organ involvement have progressed. The efficacy of treatment with mesenchymal stem cells and chimeric antigen receptor T cells for SS has also been investigated. In this review, we will provide an overview of the pathogenesis of salivary gland lesions and recent trends in therapeutic approaches for SS.

## 1. Introduction

Sjögren’s syndrome (SS) is a chronic autoimmune disease that initially presents with symptoms of dry keratoconjunctivitis and parotid swelling, and accompanies other immune disorders, such as arthritis. This autoimmune disease affects exocrine glands, such as the salivary and lacrimal glands, with a female predilection [1,2,3,4,5]. SS is generally divided into primary and secondary forms. Secondary SS is associated with rheumatoid arthritis (RA), systemic lupus erythematosus (SLE), scleroderma, dermatomyositis, and mixed connective tissue disease, while primary Sjögren’s syndrome (pSS) is uncomplicated [6]. pSS is further divided into the glandular form, which is confined to the lacrimal and salivary glands, and the extraglandular form in which systemic organs, such as the lungs, kidneys, skin, blood, and peripheral nerves are involved; 30–40% of SS patients have joint, kidney, lung, and peripheral nervous system involvement [5]. Blood tests detect peripheral B-cell subsets disturbance and increase in serum B cell activating factor (BAFF), gammaglobulinemia, and the ribonucleoproteins SS-related antigen A (SSA)/Ro, and SS-related protein B (SSB)/La [7]. Among cases of pSS, 5–10% progress to the lymphoma stage and develop B-cell non-Hodgkin lymphomas, with MALT lymphoma predominantly [6,7,8,9].

The salivary glands are the site of the initial manifestation of the disease and of high incidence of lymphoma. Since the lymphocyte infiltration of exocrine glands and ectopic germinal centers (GCs) (eGC) are also observed in other affected organs of SS patients, it is important to elucidate the pathogenesis of salivary gland lesions in order to obtain a more detailed understanding of this disease [3,8,10]. A number of factors including genetic predisposition, environmental factors, sex, and viral infection are involved in the pathogenesis of SS [11]. Furthermore, detailed studies on pathogenic T and B cell subsets, the epigenetics of SS, and analyses of diverse genetic changes in lesions have progressed and knowledge is updated annually [12,13]. However, with the expansion of detailed research on SS, it is becoming increasingly difficult to understand the overall pathogenesis. Regarding treatment, clinical studies on effects of biologic therapies targeting specific subsets of immune cells, particularly B cells and plasma cells (PC), are underway in patients with systemic manifestations [14,15]. Basic knowledge of the mechanisms and therapeutic targets is essential to better understand and utilize the expanding range of therapeutic agents. In this review, we provide an outline of the overall process of salivary gland lesion formation in terms of autoantigen-mediated target cell injury by T cells, as well as trends in therapies to promote salivary secretion and prevent the progression of systemic manifestations.

## 2. Changes in the Salivary Gland Histology in SS

The salivary glands are divided into the major salivary glands, i.e., parotid, submandibular, and sublingual glands, and the minor salivary glands (MSG) scattered throughout the oral mucosa. The salivary gland epithelium is mainly composed of acinar, myoepithelial, and ductal cells (Figure 1A). Acinar cells are surrounded by myoepithelial cells, produce saliva, and secrete it into the lumen. The secretion of saliva is facilitated by the action of myoepithelial cells and is controlled by muscarinic parasympathetic nerves. Excretory ducts are further divided into intercalated ducts, striated ducts, and terminal excretory ducts. In terminal excretory ducts, the epithelium is multilayered with columnar cells and becomes a multilayered squamous cell epithelium as it approaches the oral cavity. Intercalated ducts are also surrounded by myoepithelial cells. The basal cell layer that surrounds striated ductal cells is presumed to contain salivary gland stem cells [8,16,17].

Salivary gland lesions in SS patients are characterized by dense periductal cellular infiltration and the loss of acini [18] (Figure 1(B1,B2)). In microscopic examinations of the salivary glands of SS patients, a periductal lymphocytic infiltrate of 50 or more cells per 4 mm^2^ of tissue is counted as one focus, which is a histological marker of diagnostic criterion in SS [19,20]. When SS progresses, the number of foci with periductal cellular infiltration increases, and the percentage of B cells in the focus gradually increases. In addition, structures similar to the GCs of lymphoid tissue, eGCs, are formed in the salivary glands and autoantibodies are produced in these lymphoid structures (Figure 1C). Following reductions in acinar cells and ductal cells, the space is often replaced by fibrous and adipose tissues [20,21,22]. When striated ductal cells are lost, terminal excretory ducts with a larger diameter will become dominant in the histology of inflamed salivary glands. Electron microscopy of the salivary glands in aggravated pSS revealed atrophic acini, shrunken nuclei, and swollen organelles. Lipid droplets were also observed in the mucous droplets of acinar cells. Ductal structures begin to atrophy and become distorted. Nuclear pyknosis, fat infiltration in the cytoplasm, and swelling of the intracellular organelles were also observed [23]. Although it has not yet been proven, the main autoimmune targets of SS are presumed to be acinar, intercalated, and striated duct epithelial cells.

Lymphoepithelial lesions develop in the striated ducts and are characteristic ductal lesions of pSS (Figure 1D). They form adjacent to periductal lymphocyte infiltrates and consist of hyperplasic basal cell population of the ducts and infiltrating lymphocytes surrounded by B cells [24]. When acinar and ductal cells are targeted by infiltrating lymphocytes, basal cells may avoid the lymphocyte attack and proliferate, resulting in the epithelial hyperplasia of lymphoepithelial lesions.

## 3. Mechanisms Underlying the Formation of Salivary Gland Lesions in pSS

In the salivary glands of SS patients, autoantigens are recognized by antigen-presenting cells (APCs), such as dendritic cells (DCs), resulting in the activation of T cells and B cells, cytokine production, autoantibody production by B cells, the formation of eGCs, and the destruction of salivary gland epithelial cells [25,26]. We herein described the background of SS, and the functions of DCs, CD8^+^ T cells, CD4^+^ T cells, and B cells in the salivary gland lesions of SS.

### 3.1. Background of the Development of Salivary Gland Lesions

Genetic and environmental factors are involved in autoimmune diseases. Genetically, the HLA type is considered to be important [11]. Since autoimmune diseases occur in 12–15% of systemic monozygotic twins and 3–8% of dizygotic twins, environmental factors other than genetic factors have been suggested to play a pivotal role in autoimmune diseases [27,28,29]; however, this has not yet been demonstrated in SS. Genome-wide association studies identified HLA and non-HLA susceptibility genes in SS, and non-HLA genes include STAT4, C-X-C chemokine receptor (CXCR) 5 (CXCR5), TNIP1, GTF21, TNFAIP3, PTPN22, IRF5, IL21A, BLK, BLK-FAM167A, BAFF, and EBF. The HLA-genes associated with an increased risk of SS include MHC class I-encoded HLA-B8 and MHC class II-encoded HLA-DR3, HLA-DRB1, and HLA-DQB1 [30,31,32,33,34,35,36]. Khatri et al. [37] identified Cd247, NAB1, PTTG1-MIR146A, PRDM1-ATG5, TNFAIP3, XKR6, MAPT-CRHR1, PROR-CHMP6-BAIAP6, TYK2, and SYNGR1 as genes associated with the onset of SS in European ancestry.

Environmental factors for SS include infection, non-tuberculous mycobacteria, estrogen, vitamin D, stress, smoking, and silicone breast implants [36]. Viruses are considered infectious risk factors for the development of SS, among which Epstein–Barr virus (EBV), cytomegalovirus (CMV), human T-cell leukemia virus type 1, human herpesvirus 6 (HHV-6), and hepatitis C virus have been examined [38,39,40,41]. EBV, CMV, and HHV-6 are DNA viruses belonging to the herpesvirus group. They remain latently in the body and are reactivated by triggers. Except for EBV, few viruses have been shown to have a causal relationship with SS.

EBV is a gamma herpesvirus that persistently infects B cells for life and antibodies have been detected in more than 90% of humans [42,43]. EBV initially infects oral epithelial cells, which in turn results in the infection of naive B cells in the mucosa, followed by a latent infection in B cells. The latent infection of EBV in B cells is divided into Latency III, II, I, and 0, which is based on the number of genes expressed [44], with Latency 0 being the least expressed mode of infection; the genes expressed are limited to EBV-encoded small RNAs. In the recurrence of EBV, the signal for B cells to differentiate into PC or a number of stimuli, such as transforming growth factor β (TGF-β), anti-immunoglobulin, hypoxia, reactive oxygen species, and temperature shifts, may activate the virus to enter a lytic-abortive cycle [43]. Viruses reactivated from B cells infect the basal cells of the epithelium of the oral mucosa and when differentiated, these cells produce infectious viruses, shedding them into the saliva [29,45]. Lytic cycle proteins are expressed upon EBV reactivation, among which BHRF1 and BALF1, two viral orthologues of cellular Bcl-2, display anti-apoptotic functions and protect infected cells from cell death signals. BCRF1 encodes viral interleukin (IL)-10 and BARP1, which contribute to reduced natural killer (NK) and CTL responses as well as the suppression of T cell activity by inhibiting IFN-γ. BILF1, BGLF5, and BNLF2a deregulate the HLA pathway to evade the clearance of EBV-infected cells. Zta encoded by BZLF1 has been shown to directly or indirectly increase the levels of IL-13, IL-8, the CXC chemokine GRO, CCL4, IL-6, and IL-10 [46].

In a case-control study on SS, the presence of anti-SSA/Ro correlated with EBV-associated antibodies (the IgG anti-EBV early antigen and IgG anti-EBV capsid antigen). Anti-SSB/La has also been associated with the IgG anti-EBV early antigen [47,48]. Therefore, EBV is not only involved in the destruction of salivary gland cells and presentation of autoantigens to APCs at the onset of SS, it may also induce the expression of EBV lytic genes by reactivation and support immune evasion by and the proliferation of autoreactive B cells, contributing to the development and maintenance of the eGCs of the salivary glands in SS.

### 3.2. Role of DCs in Salivary Glands and Lymph Nodes

When salivary gland cells are damaged by inflammatory responses to viral infection, bacterial infection, or mechanical injury, released autoantigens are taken up by APCs, undergo partial proteasomal degradation to peptides, and are presented to T cells by the major histocompatibility complex (MHC) of APCs. DCs are highly specialized APCs that present extracellular antigens in the context of MHC class I (MHC I) molecules to CD8^+^ T cells and of MHC class II (MHC II) molecules to CD4^+^ T cells, these antigens are recognized by T cell receptors (TCR) on each T cell. The former process, referred to as cross-presentation, is important for the induction of immune responses against viruses and bacteria that do not primarily infect DCs [49,50,51]. Two cross-presentation pathways that enable HMC I molecules to be loaded with exogenous antigens have been described. When contained in the endosomal compartment, antigens are shuttled to the cytoplasm, where they are initially degraded by the proteasome. Thereafter, the resulting peptides are loaded into the endoplasmic reticulum (ER) lumen via the transporter associated antigen processing (TAP) protein, and are then incorporated into the MHC I protein complex by the chaperone tapasin [52]. Alternatively, lysosomal proteases such as cathepsin S denature the exogenous antigen in the acidic compartment, in which peptides are loaded to intra-endosomal MHC I molecules. In MHC II-restricted antigen presentation, the incorporated antigen is degraded in the endo/lysosomal compartments by proteases such as cathepsin S and antigen-derived peptides are loaded onto MHC II.

There are three types of human DCs: plasmacytoid DC (pDC), myeloid/conventional DC1 (cDC1), and myeloid/conventional DC2 (cDC2) [25,26,53]. cDC1 cells are highly capable of antigen cross-presentation to CD8^+^ T cells via MHC I. In addition, other DC subunits, including certain cDC2 subsets, induce cross-presentation of antigens [53]. Jog et al. [42] proposed a model in an article overviewing EBV and autoimmune responses in SLE. In this model, the anti-EBV antibody forms immune complexes by interacting with autoantigens, which have a common epitope with a specific viral antigen. These immunocomplexes are taken up by cDC via Fc receptor (FcR) [26,54] (Figure 2). The antigens are presumed to undergo intracellular degradation for MHC I-restricted antigen presentation to CD8^+^ T cells. Antibodies against different regions of the EBNA-1 protein have been shown to cross-react with the SLE autoantigens SmB, SmD, and Ro. EBER1 and EBER2 (RNA encoded by EBV) are bound to proteins and can be specifically precipitated by anti-La antibodies [38,55]. These antibodies (anti-Ro and anti-La antibodies) are the ones detected in SS patients. Thus, this model explaining molecular mimicry and epitope scattering in autoimmunity (SLE) [42] may be applicable to SS.

Abnormalities in the interferon (IFN) system occur in SS [56,57,58]. Previous studies on the peripheral blood and MSG of SS patients demonstrated that the type I IFN signature was predominant at the peripheral blood level, while the type II signature was predominant in the salivary glands of SS patients [59]. Among DCs, the pDC subset is considered to be the major producer of IFN-α, while Th1 and NK cells are the major producers of IFN-γ [56]. pDC constitutively expresses endosomal Toll-like receptors (TLR) 7 and 9, through which they recognize self-nucelic acids as well as viral RNA and DNA. The triggering of these TLRs leads to the production of type I IFNs or pro-inflammatory cytokines [60,61] (Figure 2)

### 3.3. Role of CD4^+^ Th, CD8^+^ T Cells in Salivary Glands

Previous studies on infiltrating cells in the salivary gland tissues of the SS detected a high number of CD4^+^ T cells; however, CD8^+^ T cells were also found among the infiltrating cells [62,63,64,65]. A recent quantitative study on MSG of the lips identified CD8^+^ T cells as the most abundant infiltrating cell type [66]. CD8^+^ effector cells, CD8^+^ cytotoxic T lymphocytes (CTL), play a major role in the direct killing of target cells, including virus-infected cells and tumor cells [67]. Since salivary gland epithelial cells express MHC class I, CD8^+^ CTL that target autoantigens defined by MHC I directly induce apoptosis in salivary gland epithelial cells though the secretion of IFN-γ, granzyme, perforin, and tumor necrosis factor (TNF)-α [68,69] (Figure 3). In addition, CD8^+^ CTLs may express Fas ligand (FasL) or secreting TNF-α, which binds to FAS and TNF receptors expressed on the surface of epithelial cells, respectively, and induces apoptosis through caspase signaling pathways [70]. CTLs also produce various cytokines such as IFN-γ, IL-4, and IL-17. The expression of C-X-C motif chemokine ligand (CXCL) 9 (CXCL9), CXCL10, and CXCL11 was found to be upregulated by IFN-γ in salivary gland epithelial cells and CXCR3 for these IFN-γ-inducible ligands was expressed on effector T cells. These salivary gland-derived cytokines and chemokines may stimulate the accumulation of CD8^+^ CTLs in the salivary glands and contribute to the exacerbation of SS lesions [71,72].

Antigen presentation by DCs causes naïve CD4^+^ T cells to differentiate into a number of T cell subsets, including T helper 1 (Th1) cells, T helper 2 (Th2) cells, T helper 17 (Th17) cells, regulatory T cells (Tregs), T follicular helper (Tfh) cells, and a few other T helper subsets, which are characterized by a series of characteristic transcription factors and a signature-specific cytokines [73,74,75] (Figure 2). Each subset produces specific cytokines to promote immune responses: Th1 cells produce IL-2, IFN-γ, and macrophage colony-stimulating factor (M-CSF); Th2 cells produce IL-4, IL-5, B cell activating factor (BAFF), and mast cell growth factor (MCGF); Th17 cells produces IL-17 and IL-22 [76]. Tfh cells secrete IL-21 and contribute to B cell maturation. Although SS has historically been regarded as a Th1-derived disease, other subsets of CD4^+^ T cells, such as Tfh and Th17, also contribute to this disease. The expression of the proinflammatory cytokines, interleukin (IL)-1, TNF-α, and IL-6 was previously shown to be increased in SS patients [77]. IL-6 produced by salivary gland epithelial cells plays important roles in T helper cell differentiation, GC formation, and PC formation. The binding of IL-6 to specific receptors results in the homodimer formation of its receptor component, g130, which in turn activates Janus kinase 1 (JAK1) and phosphorylates signal transducer and activator of transcription 3 (STAT3). STAT3 is required for the transduction of cytokine signals to the nucleus, cell proliferation, and anti-apoptotic signaling. The JAK1/STAT3 pathway is also involved in the expression of IFN-γ in Th1 cells and IL-17 in Th17 cells [78,79,80].

DCs secrete the cytokines TGF-β and IL-23 and promote the differentiation of Th17 cells [81]. IL-1β produced by the ductal epithelial cells of salivary glands is also required for Th17 polarization. Activated Th17 cells increase the proinflammatory cytokines IL-6 and TNF in inflammatory exocrine glands. IL-17 produced by Th17 induces the secretion of matrix metalloproteinase-1 (MMP-1) and -3 (MMP-3) from the glands, leading to the degradation of the extracellular matrix and glandular cell destruction [82]. IL-17 and IL-21 derived from Th17 also stimulate B cell receptors (BCR), support an isotype class switch, affect B cell trafficking within GCs, form eGCs, and support autoantibody formation [83,84].

Cytotoxic CD4^+^ T cells (CD4^+^ CTLs) have been detected in humans and mice. Unlike CD8^+^ T cells, which recognize MHC I-defined antigens, CD4^+^ CTLs recognize antigen peptides defined by MHC II to exhibit antigen-specific cytotoxicity [85,86,87]. Their numbers increase in chronic viral infections in which CD4^+^ T cells are repeatedly stimulated. CD4^+^ CTLs are generally important under conditions where CD8^+^ CTLs are depleted or MHC I is downregulated by viral evasion mechanisms [75]. MHC II is normally expressed on classical APCs (cAPCs) such as DCs, macrophages, B cells, and thymic epithelial cells, but is also expressed on non-APCs in response to IFN signaling. Therefore, Th1-derived IFN-γ has been suggested to induce the expression of MHC II in salivary gland epithelial cells. As a result, CD4^+^ CTLs target autoantigens defined by MHC II on salivary gland epithelial cells and exert their cytotoxicity (Figure 3). Antigen-specific CD4^+^ CTLs have been implicated in the pathogenesis of autoimmune diseases such as multiple sclerosis (MS), RA, and SLE [88]. In systemic sclerosis, where HLA-DR-expressing endothelial cells are targeted, CD4^+^ CTLs have been implicated in the induction of apoptosis in endothelial cells [89]. The single-cell RNA sequencing of peripheral blood mononuclear cells from pSS patients identified CD4^+^ CTLs as a significantly expanded subpopulation, suggesting the involvement of CD4^+^ CTLs in the pathogenesis of pSS [90].

### 3.4. Role of Tregs

Treg-mediated peripheral immune tolerance suppresses effector T cells and immune-mediated tissue damage. Tregs are CD4^+^ T cells that highly express CD25, an IL-2 receptor, and are recognized as a subtype of the CD4^+^ T cell lineage by the identification of the master transcription factor forkhead box protein 3 (Foxp3) [91,92]. Tregs are less responsive to TCR stimulation, but express the surface markers cytotoxic T-lymphocyte associated antigen-4 (CTLA-4), glucocorticoid-induced TNF receptor family-related protein (GITR), programmed cell death-1 (PD-1), and PD ligand 1 (PD-L1) as activating effector T cells. They are divided into two subsets: naturally occurring (nTregs) and induced Tregs (iTregs) [93,94]. Foxp3^+^ nTregs are produced in the thymus and express Foxp3 and high levels of CD25. iTregs (or adaptive Tregs) are derived from CD4^+^Foxp3^−^ naive T cells in the periphery. TGF-β and IL-2 are required for this conversion. The constitutively high expression of CD25 on Tregs allows for the efficient absorption of IL-2 and deprives it from T cells, leading to apoptosis in effector T cells [95,96]. Alternatively, Tregs inhibit IL-2 mRNA transcription in Foxp3^−^ effector T cells [97].

Another mechanism by which Tregs affect immune responses is through their interaction with DCs. Antigen-experienced effector T cells are placed in a quiescent state by DCs in the absence of Tregs. Since DCs are affected by their relationship with Tregs, the addition of Tregs reduces the expression of co-stimulatory molecules on DCs and then inhibits the interaction between DCs and effector T cells. CTLA-4 of Tregs reduces the expression of CD80/CD86 on DCs, suppresses the activation of DCs, and makes them more immune tolerant. The selective deletion of CTLA-4 in Tregs may cause systemic autoimmunity and fatal disease but did not affect the development of Tregs or convert them towards pathogenicity [98]. As an auxiliary regulatory mechanism, Tregs exert their immunosuppressive function by producing the cytokines IL-10, TGF-β, and IL-35 [94,99,100].

### 3.5. Role of B Cells and Formation of eGCs

B-cell activation is key to the pathogenesis of SS because the overactivation of B cells results in hypergammaglobulinemia and the production of the anti-ribonucleoprotein antibodies, anti-SSA/Ro, and anti-SSB/La antibodies [4,101,102] (Figure 3). Two pathways activate naive B cells: the T cell-dependent pathway to produce PC and IgM, and T cell-independent pathways to produce PC, memory B cells, IgG, IgA, and IgE [102]. In T cell-independent activation, antigen-associating nucleic acids, e.g., Ro ribonucleoproteins, provide a dual signal that stimulates BCR and ligates endosomal TLRs. In T cell-dependent activation [103,104], B cells recognize homologous autoantigens via BCR, either by directly binding soluble antigens or by binding antigens presented on the surface of DCs and macrophages [105,106]. Thereafter, they present autoantigens defined by MHC II to CD4^+^ Tfh cells, which is essential for B cell activation and GC formation [107].

TLRs are not only capable of pathogen recognition, they also recognize autoantigens, particularly those released from apoptotic cells and may be enhanced by type I IFN [108]. Type I IFN indirectly affects B cells via the induction of BAFF secretion by various types of cells. BAFF belongs to the TNF family and is produced by monocytes, macrophages, and DCs; however, in patients with pSS, it is also secreted by T cells, B cells, and salivary gland epithelial cells, the targets of autoimmune responses in SS [109,110]. BAFF transgenic mice have been shown to develop the pathological features of SLE and pSS. The BAFF axis has two ligands, BAFF itself and a proliferation-inducing ligand, to which there are three receptors with different affinities, BCMA, TACI, and BR3 [111,112]. BAFF binding to the BAFF receptor on B cells contributes to the formation of GCs, autoreactive B cell survival, and their localization in follicle/marginal zone niches in the salivary glands of pSS patients [113] (Figure 3). BCR signaling requires phophatidyl-inositol 3-kinase delta (PI3Kδ) isoform and Bruton’s tyrosine kinase (BTK) [114].

eGCs that promote the expansion of antigen-specific B cells are important in the pathogenesis of SS as well as other systemic autoimmune diseases [115]; 10–30% of patients with pSS are reported to develop this lymphoid structure [2]. GCs comprise well-defined inflammatory mononuclear cells, particularly Tfh cells, GC B cells, the follicular DC network, and endothelial venules [116,117]. The physical relationship between Tfh and B cells is established by co-stimulatory molecules such as CD28-CD80/86, inducible T cell co-stimulator (ICOS)-ICOS ligand (ICOSL), CD40-CD40 ligand (CD40L), and PD-1-PD-L1. Tfh secretes IL-4 and IL-21 to induce the differentiation of B cells into memory B cells and PC [118]. The dark zone of GCs is the site of GC B cell proliferation and somatic hypermutation, forming centroblasts. Centroblasts that express CXCR5 instead of CXCR4 follow a CXCL13 gradient to enter the light zone of GCs as centrocytes. These centrocytes then undergo positive sorting by Tfh and follicular DCs and memory B cells and PC are produced. In pSS patients with systemic manifestations, the percentage of Tfh cells in peripheral blood and salivary glands correlates well with the lymphofocus score, B cell activation indices, and disease activity [119,120]. The formation of GCs in SS is also considered to be a marker of a high focus score, anti-Ro/La positivity, the presence of extraglandular lesions, and lymphomagenesis [2,112].

Follicular regulatory T (Tfr) cells have been identified as a CD4^+^ T cell that shares characteristics with Tfh cells and Tregs and regulates GC responses by suppressing Tfh and B cells [121]. This subset originates from thymus-derived Tregs rather than naive T cells; Tfr cells express BCL6, CXCR5, and PD-1, similar to Tfh cells, and additionally express CTLA-4, GITR, DC25, and IL-10 [122,123,124]. Therefore, Tfr cells resemble Tfh cells because of follicular localization, but functionally resemble Treg. CTLA-4 expressed in Tfr cells acts to block Tfh and B cell CD80/86 co-stimulation signals [115]. Therefore, Tfr cells inhibit B cell metabolism primarily by reducing glucose uptake and utilization, thereby reducing the expression of B cell effector molecules. On the other hand, Tfr cells inhibit GC responses by secreting the inhibitory cytokines IL-10 and TGF-β and reducing the secretion of IL-21 and IL-4 by Tfh cells [125,126,127].

In T cell-dependent B cell activation, autoantigens are taken up by B cells via BCR, degraded to peptides, and then presented with an MHC II-defined epitope. When B cells associate with Tfh cells, which recognize the same epitope, they become activated and differentiate into PC producing autoantibodies and memory B cells [42]. This autoimmune response is not limited to the epitope that cross-reacts with the viral protein with structural similarity [42,128], but expands to several different epitopes, through the process of epitope spreading, and produces autoantibodies that target additional antigens (Figure 4). In SS patients, many autoantibodies have been reported to date. Target autoantigens include aquaporin-5 (AQP5), α-fodrin, carbonic anhydrase 6, centromere protein, nuclear autoantigen 14 kDa, mouse double minute 2, muscarinic acetylcholine receptor 3 (M3R), parotid secretory protein, salivary protein-1, SSA/Ro, SSB/La, and tripartite motif-containing protein 38 [129,130,131,132,133,134].

B cell hyperactivity is also indicated by the presence of clonal proliferation of B cells in the lip and parotid glands of patients with pSS shown [135,136]. Current studies have identified several novel B cell subsets with multiple functions in pSS. These include autoreactive age-related B-cells, plasma cells with enhanced autoantibody production, and a tissue-remnant Fc receptor-like 4 (FcRL4)^+^ B-cell subset with enhanced inflammatory cytokine production [7]. Several genes associated with lymphoma have been found to be upregulated in FcRL4^+^ B-cells [137]. This reinforces the hypothesis that neoplastic MALT B-cells may arise from glandular FcRL4^+^ B-cells by clonal proliferation.

### 3.6. Role of PD-1 and the SS-like Pathology Induced by Anti-PD-1 Antibodies

PD-1 is mainly expressed on activated T cells and functions as a brake of T cell activation binding to PD-1L/PD-L2, the ligands expressed in peripheral tissues. Since T cells, B cells, macrophages, and some DCs express both PD-L1 and PD-1, a bidirectional relationship is formed in these cells [138,139]. Upon ligand binding, PD-1 is phosphorylated by the protein tyrosine kinase Lck, leading to the recruitment of the tyrosine phosphatase Shp2 and the subsequent dephosphorylation of CD28. As a result, TCR/CD28 signaling and subsequent T cell activation signaling are suppressed [140,141]. PD-1 also modifies T cell membrane-proximal signaling events and inhibits PI3K activity and downstream Akt activation. Since these enzyme activities are key in glucose transport and glycolysis, the suppression of these signals by PD-1 inhibits cellular bioenergetics [142,143].

In the tumor microenvironment, the PD-1 ligand, PD-L1, is upregulated by IFN-γ and other cytokines derived from infiltrating activated T cells and induces resistance to tumor T-cell immunity [144]. Therefore, the antibody blockade of PD-1/PD-L1 may restore these exhausted T cells and enhance anti-tumor immunity. PD-1 is highly expressed on Tregs and plays an inhibitory role in antitumor immunity; therefore, the blockade of the PD-1/PD-L1 interaction may promote antitumor responses by impairing the suppressive activity of Tregs [145]. The antitumor effects of immune checkpoint inhibitors (ICIs), antibodies, that block PD-1 or PD-L1, were found to be enhanced by redirecting the function of tumor-associated macrophages, the NK cell-DC axis in the tumor environment [146,147]. On the other hand, C57BL/6 mice lacking PD-1 developed a late-onset SLE-like pathology characterized by autoantibodies and mild glomerulonephritis. A PD-1 deficiency has been shown to accelerate autoimmunity in an autoimmune-prone background, indicating a role for PD-1 in the induction and maintenance of tolerance [148,149]. Therefore, the treatment of malignancies with ICIs targeting PD-1/PD-L1 may induce immune-related adverse events (AEs). In cancer patients treated with monoclonal antibodies against CTLA-4, or PD-1, inflammatory diseases including colitis, pneumonitis, arthritis, inflammatory myopathy, vasculitis, nephritis and sialadenitis, resembling pSS, occurred at a frequency of approximately 60% [150,151,152,153]; xerostomia and dry eye due to excretory dysfunction was observed in 5% of patients treated with ICIs [154]. Patients who developed xerostomia after being treated with ICIs had anti-SSA/Ro and periductal lymphocyte infiltration in MSG, and 62% met the diagnostic criteria for pSS [155]. Pringle et al. [156] reported a patient who developed dry mouth after receiving a PD-L1 checkpoint inhibitor (durvalumab) for non-small cell lung carcinoma. The patient was not able to produce any unstimulated or stimulated parotid saliva and had anti-SSA/Ro antibodies, anti-nuclear antibodies (ANA) in blood, and a positive focus score in the salivary glands. Infiltrating cells in the parotid gland were CD4^+^ T-dominant. AQP5^+^ CK7^−^ acinar cells, AQP5^−^ CK7^+^ intercalated duct cells, and AQP5^+^ CK7^+^ hybrid cells resembling intercalated duct cells, but not conventional acinar cells, were detected [156]. By blocking the PD-1/PD-L1 pathway through the administration of ICIs, anti-tumor CD8^+^ CTLs recognize autoantigens defined by HMC I on acinar and ductal epithelial cells and attack these cells, leading to the development of pSS-like disease (Figure 3). This emphasizes the important role of the PD-1/PD-L1 pathway and CTL in the pathogenesis of pSS.

## 4. Criteria for the Diagnosis and Evaluation of Disease Activity

Criteria from the American-European Consensus Group published in 2002 and the Sjögren’s International Collaborative Clinical Alliance Cohort published in 2012 list dryness, autoantibody testing, and tissue histopathology for classifying patients as having pSS. MSG biopsy and autoantibodies specific for pSS play a central role in the 2016 American College of Rheumatology (ACR)/European League Against Rheumatism (EULAR) classification criteria [4,157,158]. However, the new system, in contrast to previous ones, assigns a weighted score to each element. The ACR/EULAR criteria contain the following elements. (1) Histopathological evidence of focal lymphocytic sialadenitis with focal score ≥1 (3 points); (2) anti-SSA/Ro antibodies are positive (3 points). (3) SICCA ocular staining score ≥5 or Rose Bengal score ≥4 as assessed by the van Bijsterveld scoring system (1 point). (4) Schirmer I test ≤ 5 mm/5 min (1 point); (5) unstimulated total salivary flow ≤ 0.1 mL/min (1 point). Adoption of this new classification criterion has enabled the inclusion of patients with SS presenting with prominent extraglandular manifestations in clinical trials.

To assess treatment efficacy for pSS, current clinical trials primarily use the EULAR Sjögren’s Syndrome Disease Activity Index (ESSDAI) and the EULAR Sjögren’s Syndrome Patient Reported Index (ESSPRI) [159,160]. ESSDAI is an optimized measure to assess systemic activity and patient symptoms, with 12 domains as a multi-domain index. This includes constitution, lymphadenopathy, glandular, joint, skin, lung, kidney, muscle, peripheral nervous system (PNS), central nervous system (CNS), hematological, and biological domains. Weight coefficients are assigned to each item. Each item has a unique weight factor, is multiplied by the degree of activity (0–3) and is scored between 0 and 123. In clinical trials, the ESSDAI is considered the gold standard for assessing global disease activity. ESSDAI defines the minimum clinically significant difference as a decrease in at least 3 scores. In evaluations of the efficacy of an investigational drug, an ESSDAI of at least 5 with at least intermediate activity is a suitable candidate; an ESSDAI of 13 or higher is associated with an additional risk of developing lymphoma during the course of the disease.

The second tool is ESSPRI [161]. It is designed to assess a patient’s subjective symptoms and is defined as the average score obtained by patients self-rating three questions on an 11-point scale: dryness, pain, and fatigue. ESSDAI and ESSPRI will be included in clinical trials as criteria for evaluating new treatments for pSS; however, dryness may only be assessed by the presence or absence of glandular swelling of the salivary glands and by a patient’s subjective perception of dryness. Therefore, the evaluation of treatment effects on salivary gland function requires separate measurements of unstimulated and stimulated salivary flow.

In subsequent studies, a high activity level (ESSDAI ≥ 14), a medium activity level (5 ≤ ESSDAI ≤ 13), and a low activity level (ESSDAI < 5) were defined; minimum clinically important improvement (MCII) in ESSDAI was defined as at least a 3-point improvement. The patient-acceptable symptom state estimate was defined as an ESSPRI < 5 points and MCII as a decrease of at least one point or 15% [162]. Ultrasound of the salivary glands is being evaluated as a new diagnostic modality [163].

## 5. Treatment of Xerostomia

Patients with pSS have difficulty chewing and swallowing dry foods and speaking due to decreased saliva production. The treatment of sicca is essential for improving the quality of life of pSS patients. Patients with residual salivary gland function with moderate to severe xerostomia need to be administered muscarinic agonists to stimulate salivation. Evidence-based treatment options are limited, pilocarpine and cevimeline have been shown to stimulate salivation and improve subjective symptoms [164,165,166].

### 5.1. Pilocarpine

Pilocarpine is a parasympathomimetic agent that functions as a muscarinic agonist. It binds to M3R on exocrine glands and stimulates exocrine secretion, such as diaphoresis, salivation, and lacrimation, and gastric and pancreatic secretion. When mice were continuously treated with pilocarpine, the expression of the M3R gene and protein was upregulated [167]. In randomized trials, pilocarpine was tolerated well and significantly improved salivation and xerostomia in SS patients [165,166]; 61% of patients showed general improvement; however, not all pSS patients responded well. Saligren^®^ is a pilocarpine hydrochloride that is used to treat SS and radiotherapy-induced xerostomia; 5 mg four times a day is recommended for SS. Although its administration has been shown to attenuate symptoms, such as dysphagia and clinical eating, side effects, including sweating and excessive lacrimation, have been reported in a volume-dependent manner [168,169]. According to Noaiseh et al. [170], the side effects of pilocarpine included sweating, nausea, dyspepsia/vomiting, flushing/hot flashes, paresthesia, myalgias, headaches, and rash.

Watanabe et al. [171] created a low-dose pilocarpine formulation of a pilocarpine/sodium alginate solution to reduce the dose and side effects of pilocarpine. This pilocarpine preparation was spread over the oral mucosa without swallowing, and excess fluid was spit out after 5 min and eating or drinking was not allowed for 60 min. Treatment with this preparation was performed three times a day. Salivary flow and xerostomia were significantly improved by this treatment. The only side effect was sweating. A lyophilized hyaluronic acid containing pilocarpine has also been developed [172]. In an animal study, the drug was applied to the oral mucosa. The concentration of the drug increased in submucosal tissue, but not in the blood. In comparison with the intragastric administration of pilocarpine, the same salivary secretion-promoting effects were obtained even though the blood concentration did not increase. These localized administration products are required to reduce failure rates due to the side effects of pilocarpine.

### 5.2. Cevimeline

Cevimeline is a quinuclidine derivative of acetylcholine that directly stimulates M1R and M3R in the salivary glands [173] and received FDA approval in 2000 [174,175]. Animal and human studies demonstrated that the duration of cevimeline-induced salivation was twice as long as that of pilocarpine [175]. Loy et al. [176] confirmed that the administration to humans increased salivary gland secretion from the buccal glands based on findings obtained using optical and electron microscopies. The recommended dose of cevimeline for xerostomia in SS patients is 30 mg orally three times a day. According to Noaiseh et al. [170], frequently observed side effects were sweating, nausea/dyspepsia/vomiting, flushing/hot flashes, headaches, and breast swelling. In SS patients, the failure rates of pilocarpine and cevimeline were 47% and 27%, respectively, among first time users. Severe sweating was the most frequent side effect leading to the cession of therapy and occurred more frequently in pilocarpine users (25%) than in cevimeline users (11%).

### 5.3. Sialendoscopy

Salivary gland endoscopy is an endoscopic diagnostic instrument for the major salivary glands that is used to treat stricture, mucous plugs, and salivary stone-related chronic obstructive disease [177,178]. Salivary gland endoscopic lavage with saline has been reported to promote salivation [179]. The combination of salivary gland endoscopy and triamcinolone acetonide (TA) improved efficacy over saline alone [178]. When the salivary gland endoscopic cleaning procedure was evaluated in SS patients after 60 weeks, salivary outflow was higher than pre-treatment levels [180]. A limitation of this treatment was the difficulty associated with identifying the orifice of the salivary excretory duct or gland endoscopy for Wharton’s ducts of the submandibular gland is more complicated than that for the parotid gland [181]. Many salivary gland imaging studies have shown stenosis in each branch of the ductal system in SS patients. Abnormalities in the main ducts were mainly dilation, not stenosis. Therefore, salivary gland endoscopic treatment is indicated for stenosis in the definitive ducts, and is not always recommended for SS patients without stenosis because of the complications associated with this treatment. Du et al. [182] demonstrated that the use of saline or TA in salivary gland duct irrigation increased salivation over baseline, with better outcomes being achieved in patients with non-severe xerostomia. Therefore, the cleaning procedure of the ducts without salivary gland endoscopy is effective for SS patients. The less invasive nature of this treatment will allow its repetition for this long-lasting disease.

## 6. Treatment with Anti-Rheumatic Drugs

Non-steroidal anti-inflammatory drugs, glucocorticoids, immunosuppressive agents, and biological inhibitors are currently used in the systemic treatment of SS [183]. A combination of these agents is required when multiple organs are involved [184,185,186]. Only a few randomized controlled trials treated pSS with conventional disease-modifying anti-rheumatic drugs (DMARDs). Therefore, DMARDs have been used for SS based on their effects on other autoimmune diseases, such as SLE and RA [187]. Conventional drugs for RA are methotrexate (MTX), sulfasalazine, azathioprine, cyclosporine A (CyA), leflunomide, and hydroxychloroquine (HCQ) [188,189,190,191,192]. A pilot study on the efficacy and safety of MTX, administered at 0.2 mg/kg body weight once a week for one year, showed that subjective parameters, such as dryness of the eyes and mouth, were ameliorated; no improvements were observed in objective measures of dryness of the eyes and mouth (Schirmer’s test, the saliva flow rate). Elevated IgG and ERS observed at baseline persisted [190]. A double blind, placebo-controlled trial on CyA in patients with pSS was conducted at 5 mg/kg body weight for 6 months, which significantly attenuated xerostomia in CyA-treated patients, but not in controls. None of the other subjective and objective parameters significantly changed after the treatment in either group. Histological deterioration was noted in the placebo group, while histological lesions remained unchanged in the CyA group after 6 months of treatment, suggesting that CyA retarded disease progression. In a double-blind, placebo-controlled trial on pSS patients with low disease activity, HCQ did not attenuate the most common symptoms, dryness of the eyes and mouth, arthralgia/general pain, and fatigue until week 24. Slight decreases were observed in ESR, IgM, and IgG levels [191].

### Iguratimod

Iguratimod, a small molecule compound with diverse immunomodulatory activities, is a promising and widely used agent in the treatment of rheumatic diseases. Previous studies demonstrated that iguratimod was an effective therapeutic agent for pulmonary fibrosis and osteoporosis [193]. Iguratimod has been shown to inhibit B cell function by reducing the production of immunoglobulins and various inflammatory cytokines, such as IL-1, IL-6, IL-8, and TNF [194]. In a meta-analysis study on randomized trials combined with other therapies for SS, iguratimod improved ESSDAI, ESSPRI, and Schirmer’s test scores and decreased ESR and RF levels and B cell counts without increasing the frequency of AEs. The recommended duration was at least 12 weeks [195].

## 7. Biological Therapy for Molecular Targets

In SS patients, proinflammatory cytokines, including type I and type II IFNs, are overexpressed in salivary gland tissues and peripheral blood, and the Janus kinase (JAK)/STAT pathway plays an important role in their signal transduction [196,197]. Previous studies reported that the expression of BTK was higher in the B cells of SS patients than in those of healthy controls. Additionally, spleen tyrosine kinase (SYK) and BTK have been shown to play a key role in the transduction of BCR signals [198,199]. Therefore, JAK/STAT, SYK, and BTK have potential as targets for SS therapy. Filgotinib is an oral preferential JAK-1 inhibitor that is used in the treatment of moderate to severe RA; lanraplenib is an oral inhibitor of SYK and in development for the treatment of inflammatory and autoimmune diseases; tirabrutinib, a selective inhibitor of BTK, is also under development for the treatment of B-cell malignancies and inflammatory diseases. A randomized, phase 2, double-blind, placebo-controlled study was performed to examine the effects of filgotinib, lanraplenib, and tirabrutinib and found no significant differences in ESSDAI or ESSPRI between the SS and placebo-control groups [200]. Baricitinib is a selective JAK1/2 inhibitor and has therapeutic effects in RA patients. In a pilot study, baricitinib improved symptoms of arthritis and skin rash in patients with SS [201]. PI3Kδ is involved in B cell receptor signaling. A phase 2 randomized, double-blind, placebo-controlled proof-of-concept trial of seletalisib, a potent and selective PI3Kδ inhibitor, in pSS showed a trend toward improvement in ESSDAI and ESSPRI at week 12. No significant changes in saliva and tear flow were observed [202]. Infliximab is a chimeric anti-TNF-α monoclonal antibody. A randomized, double-blind, placebo-controlled study in which patients received an infliximab infusion or placebo on weeks 0, 2, and 6 and were followed up for 22 weeks, reported no evidence of improvements in oral or systemic findings [203].

The most focused biological therapies are monoclonal antibody therapies that affect B cells and B cell homeostasis or co-stimulation molecules [204] (Figure 5). The main anti-B cell agents are an anti-CD20 antibody (rituximab), anti-BAFF antibody (belimumab), a combination of both, an anti-CD22 antibody (epratuzumab), and anti-BAFF receptor antibody (ianalumab). The blockers of co-stimulation molecules are CTLA-4 IgG (abatacept), anti-CD40 antibody (iscalimab), and anti-ICOSL antibody (prezalumab) (Table 1).

### 7.1. Rituximab

Rituximab is a chimeric IgG1 monoclonal antibody that targets CD20 expressed on B cells and is approved for use in the treatment of B-cell malignancies. It has also been applied to the treatment of autoimmune diseases including RA [205,206,207,208]. The efficacy and safety of B cell depletion with rituximab in 30 patients with pSS was examined in a randomized, double-blind, placebo-controlled trial by Meijer et al. [209]. Observations over 48 weeks revealed significant improvements in stimulated salivary secretion and various laboratory data in the rituximab group from those in the placebo group. One case developed mild serum-sickness disease. Carubbi et al. [210] performed a case-control study on 41 patients with pSS to compare rituximab with conventional DMARDs with 120-week follow-up. The treatment with rituximab resulted in a faster and more pronounced decrease in ESSDAI than the treatment with DMARDs and significant improvements in other clinical parameters, such as the dryness visual analogue scale (VAS), unstimulated saliva flow, fatigue VAS, physician global assessment VAS, and Schirmer’s test. No AEs were observed in the two groups. In contrast, Bowman et al. [211] performed a multicenter, randomized, double-blind, placebo-controlled, parallel-group trial on 133 pSS patients and found similar response rates of 36.8% and 39.8% in the placebo and rituximab groups, respectively. No significant differences were observed in AEs between the two groups. Fisher et al. [212] examined the effects of rituximab on salivary gland ultrasound scores in pSS patients. Ultrasonographers completed a 0–11 total ultrasound score (TUS), comprising the domains of echogenicity, homogeneity, glandular definition, involved glands, and hypoechoic foci size and analyzed baseline-adjusted TUS. A significant reduction was observed in TUS at weeks 16 and 48. TUS did not correlate with the total ESSDAI score, the ESSDAI glandular domain, or salivary flow rate at any time point. The use of rituximab in patients with systemically active disease may be defended given the lack of alternatives and the safety profile observed in registry and trial data [213].

### 7.2. Epratuzumab

Epratuzumab is a monoclonal antibody that targets CD22 in B cells, a type I transmembrane protein, which is expressed at low levels in immature B cells, at high levels in mature IgM^+^ IgD^+^ B cells, and absent in differentiated PC [214,215]. Steinfeld et al. [216] performed an open-label phase I/II study on 16 pSS patients. They found that 43% achieved a clinical response (at >20% improvement level) at 6 weeks, with 53%, 47%, and 67% responding at 10, 18, and 32 weeks, respectively. Significant improvements from baseline were observed in fatigue, and patient and physician assessments at 32 weeks. Mean B-cell levels decreased by 54% at 6 weeks, which persisted in subsequent evaluations, with no evidence of the onset of recovery by the final evaluation at 32 weeks. Two patients had acute infusion reactions at the time of administration.

### 7.3. Belimumab

Belimumab is a human monoclonal antibody targeting BAFF. It is approved for the treatment of SLE and active lupus nephritis [217,218]. De Vita et al. [219] conducted an open-label phase II study on pSS patients with anti-SSA or anti-SSB antibodies, systemic complications, early disease, or biomarkers of B-cell activation. pSS patients were treated with belimumab for 12 months, and its efficacy and safety were examined. The improvements observed in ESSPRI scores at week 28 continued until week 53. Responses in ESSDAI domains (glandular, lymphadenopathy, and articular) were also observed. The decreases observed in the biomarkers of B-cell activation at week 28 remained unchanged until week 52. Therefore, long-term treatment may be beneficial for pSS patients.

### 7.4. Belimumab Plus Rituximab

B cells in peripheral blood are rapidly reduced by rituximab, whereas tissue-resident CD20^+^ B cells in inflamed tissue are less responsive to the depleting effects of rituximab. The limited efficacy of rituximab in pSS may be explained by rituximab increasing serum BAFF levels, which promotes the re-emergence of autoimmune B cells during repopulation of B cells, resulting in disease relapse over time [220,221]. Therefore, sequential treatment with belimumab and rituximab is regarded as a combined therapeutic strategy for autoantibody-positive autoimmune diseases. Mariette et al. [222] conducted a randomized, phase 2 study with two drugs on 60 patients with pSS. Belimumab plus rituximab achieved the almost complete elimination of CD20^+^ B cells in MSG and the more sustained elimination of peripheral CD19^+^ B cells than their single use. After 68 weeks, the mean total ESSDAI score decreased from a baseline of 11.0 to 5.0 in the combination group (Table 1). In the placebo group, this value changed from 10.4 to 8.6. The combined treatment removed B cells more effectively than their single use. The safety profile of belimumab plus rituximab in pSS was consistent with those of the monotherapies.

### 7.5. Ianalumab

Ianalumab is a human IgG1 monoclonal antibody that targets the BAFF receptor. Dörner et al. [223] conducted a double-blind, placebo-controlled, phase 2, single-center study on 27 pSS patients at varying doses. Patients were randomized to a single infusion of ianalumab at 3 mg/kg or 10 mg/kg or a placebo. The findings obtained on ESSDAI and ESSPRI showed the therapeutic effects of ianalumab. The rapid and prolonged depletion of B cells occurred after a single dose. Serum Ig light chains decreased and returned to baseline at the end of the study. In the high-dose group, several clinical outcomes persisted after the end of treatment. Side effects were generally mild to moderate infusion reactions within 24 h of administration of ianalumab. Overall, the single-dose regimen exerted a cooperative and sustained B-cell-depleting effect, which may be beneficial in the treatment of patients with pSS without major side effects. Bowman et al. [224] conducted a randomized, double-blind, placebo-controlled, phase 2b dose-finding trial on 150 patients with moderate to severe pSS. Ianalumab was subcutaneously administered at a dose of 5, 50, or 300 mg. The ESSDAI score was lower in the ianalumab-treated groups than at baseline (Table 1), with a maximal score change from baseline being observed in the ianalumab 300 mg group. There were four serious AEs in three patients. Of these, treatment-related pneumonia, and gastroenteritis were detected in the placebo group, and appendicitis and tubo-ovarian abscesses in one patient in the ianalumab 50 mg group. Overall, the drug was tolerated well and safe, with no increase in infections. This is the first large, randomized, controlled trial on pSS that met its primary endpoint.

### 7.6. Abatacept

Abatacept is a fusion protein of the Fc region of IgG1 and the extracellular domain of CTLA4, an inhibitory checkpoint molecule that antagonizes CD28 in its binding to CD80/CD86 on antigen-presenting cells and CD28 on T cells [225,226]. It inhibits the T cell activation step, which is key in the pathogenicity of pSS. Abatacept was proven to be effective in RA patients [227]. Meiners et al. [228] conducted an open-label study on 15 patients with early active pSS. The total treatment period was 24 weeks. ESSDAI and ESSPRI scores and RF and IgG levels significantly decreased during the treatment with abatacept, e.g., at weeks 12 and 24, and returned to baseline levels at week 48. Salivary and lacrimal gland functions did not significantly change during treatment. Six patients (40%) developed mild acute AEs with dizziness and hypotension. During the treatment, 18 self-reported infections were observed in 10 patients (67%). Baer et al. [229] conducted a phase III, randomized, placebo-controlled trial of 187 patients with active pSS. Patients were subcutaneously administered abatacept or placebo weekly for 169 days. After 169 days, the adjusted mean change in ESSDAI scores from baseline was −3.2 in the abatacept group and −3.7 in the placebo group. No significant differences were noted between the two groups.

### 7.7. Iscalimab

The CD40-CD40 ligand (CD40-L) pathway has been identified as an important co-stimulatory pathway driving T cell-dependent B cell activation and humoral immune responses [230]. CD40 engagement on B cells promotes B cell activation, and proliferation and the formation of GCs. The inhibition of the CD40-CD40L pathway was previously reported to suppress the autoimmune pathology in SS models, suggesting its involvement in the formation of eGCs and antibody-producing PCs in the salivary glands of SS [231,232]. Fisher et al. [233] conducted a multicenter, randomized, double-blind, placebo-controlled, proof-of-concept study on 44 pSS patients with the anti-CD40 antibody iscalimab. In the double-blind phase, cohort 1 patients received subcutaneous iscalimab or a placebo on weeks 0, 2, 4, and 8; cohort 2 patients received intravenous iscalimab or a placebo on weeks 0, 2, 4, and 8. In week 12, the double-blind study was terminated and changed to an open-label study and patients in both cohorts received iscalimab at the same dose and via the same route for another 12 weeks. The intravenous treatment with iscalimab resulted in a mean reduction of 5.21 points in the ESSDAI score from that of the placebo. No significant differences were observed in ESSDAI scores between the groups subcutaneously treated with iscalimab and the placebo and AEs were similar.

### 7.8. Prezalumab (MED15872/AMG557)

ICOS is a co-stimulatory checkpoint molecule that is expressed on T cells, binds to ICOSL on B cells, and is involved in T cell-dependent B cell activation. The expression of ICOS and its ligand, ICOSL, together with CD28 signaling, is upregulated in the salivary glands of pSS patients, particularly in the presence of eGCs [234,235]. A non-depleting, blocking anti-ICOSL monoclonal antibody was shown to reduce IL-21, TNF-α, IL-6, and IL-8 levels in pSS salivary gland organ culture experiments [235]. Mariette et al. [236] conducted a phase 2a, randomized, placebo-controlled study on pSS patients using the anti-ICOSL antibody, prezalumab (AMG557/MEDI5872). Patients received prezalumab 210 mg subcutaneously once weekly for 3 weeks and then every 2 weeks for 9 weeks. Although RF levels were reduced in patients with active pSS, prezalumab 210 mg did not achieve consistent improvements in clinical or other biomarker measures of disease activity. The incidence of AE was 68.8% for prezalumab and 87.5% for the placebo, with no significant differences between the two groups.

**Table 1 ijms-24-11179-t001:** Clinical effects of biological therapies in Sjogren syndrome.

Drugs	Authors Year [Ref]	No. of Subjects	ESSDAI (0–123)	ESSPRI (0–10)	Oral Dryness (VAS)	Unstimulated Salivary Flow	Stimulated Salivary Flow
						(mL/min)	(mL/min)
Rituximab	Meijer et al. 2010	30				W0 W48	W0 W48
(anti-CD20)	[209]					C: 0.06 → 0.05	C: 0.7 → 0.18
						T: 0.17 → 0.18	T: 0.92 → 0.66
	Carubbi et al.	41	W0 W120		W0 W120	W0 W120	
	2013 [210]		C: 19.8 → 8.8		C: 72 → 51.8	C: 0.08 → 0.1	
			T: 20.3 → 5.2		T: 72 → 25.1	T: 0.08 → 0.4	
	Bowman et al.	133	W0 W48	W0 W48	W0 W48	W0 W48	
	2017 [211]		C: 6.0 → 4.5	C: 6.7 → 4.5	C: 77.3 → 70.5	C: 0.08 → 0.04	
			T: 5.3 → 6.3	T: 6.4 → 6.3	T: 75.3 → 66.4	T: 0.08 → 0.07	
Epratuzumab	Steinfeld et al.	16	Physician	Patient			
(anti-CD22)	2006 [216]		assessment VAS	assessment VAS		Improvement rate	
			W0 W32	W0 W32		W18 W32	
			56 → 26	62 → 40		34% → 46%	
Belimumab	De Vita et al.	28	W0 W52	W0 W52	W28 W52		
(anti-BAFF)	2015 [219]		7 → 3	6 → 4.5	4.9 → 5.1		
Belimumab +	Mariette et al.	60	W0 W68	W0 W68		W0 W68	W0 W68
Rituximab	2022 [222]		C: 10.4 → 8.6	C:6.4 → 5.7		C: 0.12 → 0.11	C: 0.46 → 0.36
			T: 11.0 → 5.0	T: 6.0 → 5.2		T: 0.12 → 0.17	T: 0.72 → 0.9
Ianalumab	Dörner et al. 2019	27	Change from W0	Change from W0			
(anti-BAFF	[223]		W24	W24			
receptor)			C: −2	C: −0.03			
			T: −4	T: −0.3			
	Bowman et al.	150	Change from W0				
	2022 [224]		to W24 (300 mg iv)				
			−1.92				
Abatacept	Meiners et al.	15	W0 W24	W0 W24		W W24	W0 W24
(CTLA4-Ig)	2014 [228]		11 → 3	7.0 → 5.8		0.17 → 0.16	0.4 → 0.41
	Baer et al. 2021	187	D1 D169	D1 D169	D1 D169	D1 D169	D1 D169
	[229]		C: 10.1 → 6.4	C: 6.5 → 5.0	C:6.9 → 5.7	C: 0.1 → 0.03	C: 0.9 → 0.1
			T: 8.7 → 5.5	T: 6.6 → 5.3	T:7.3 → 6.9	T: 0.1 → 0.02	T: 1.1 → 0.1
Iscalimab (anti-CD40)	Fisher et al. 2020 [233]	44	Cohort 2 (10 mg/kg)				
			W0 W32	W0 W32			
			C:11 → 7	C:7 → 6.6			
			T: 11 → 2.1	T:7 → 4.6			
Prezalumab	Mariette et al.	32	W0 W49				
(MED15872/	2019 [236]		C:11.6 → 2.3				
AMG557)			T: 11.8 → 3.8				
(anti-ICOSL)							

W, week; W0, baseline; C, control (placebo) group; T, tested group.

## 8. Future Prospectives (Figure 5)

Sphingosine-1-phosphate (S1P) receptor type 1 (S1PR1) plays a major role in lymphocyte migration by binding to its ligand, S1P. High S1P concentrations in blood vessels and the lymphatic circulatory system and low S1P concentrations in the thymus and lymph nodes form a gradient that facilitates the migration of lymphocytes from peripheral lymphoid organs to the circulatory system via S1PR1. Due to the central role of S1P in lymphocyte transport, S1PR1 has been implicated in autoimmune diseases and cancer [237,238,239]. Fingolimod (FTY720) is structurally similar to sphingosine and is phosphorylated by sphingosine kinase 2. The rationale for this therapy is that phosphorylated fingolimod binds to the S1P receptor on lymphocytes, leading to its internalization and degradation. Fingolimod has been shown to effectively ameliorate disease activity in mouse models of MS, transplantation, and uveitis by preventing the migration of immunoreactive lymphocytes to target organs [240,241]. Fingolimod was approved by the US FDA in 2010 for the treatment of relapsing and remitting MS, based on the findings of a large clinical study [241,242]. In a model of viral sialadenitis, the administration of cenerimod, a potent and selective oral S1PR1 modulator, also reduced salivary gland infiltrating cells, destroyed ectopic lymphoid structures, attenuated salivary gland inflammation, and preserved organ function [243]. These S1P receptor modulators have potential as therapeutic agents for SS.

### 8.1. Mesenchymal Stem Cells (MSCs)

MSCs are pluripotent stromal cells derived from the mesoderm and ectoderm that have the ability to self-renew and differentiate into osteoblasts, adipocytes, and chondrocytes. MSCs may be isolated from most adult tissues, including bone marrow, the umbilical cord blood, adipose tissue, dental tissue, skin, and placenta [244,245,246,247]. MSCs are hypoimmunogenic because they do not express HMC II and have low MHC I expression. Previous studies reported that MSCs suppressed T and B cell proliferation at the peak and onset of disease, suggesting their immunomodulatory and anti-inflammatory effects [248,249,250,251]. The transfer of MSCs into mouse SS models was shown to induce Tregs and suppress Th1, Th17, and Tfh cell responses, thereby inhibiting autoimmune reactions and restoring salivary gland secretory function. Furthermore, MSCs may differentiate into salivary gland epithelial cells, which may be an appropriate alternative treatment to preserve the organ [252]. In a study by Xu-J et al. [253], 24 patients with SS were intravenously administrated umbilical MSCs. MSC therapy clearly reduced SS symptoms and notably decreased ESSDAI scores and VAS. The salivary gland secretion rate was also improved. No serious AEs were observed during or after treatment.

### 8.2. Chimeric Antigen Receptor T (CART) Cells

CART cells are T cells genetically engineered with four major domain receptors for immunotherapeutic purposes. The receptor consists of an antigen-recognition domain, such as a single-chain variable fragment antibody, an extracellular hinge or spacer domain, a transmembrane domain, and a co-stimulatory domain and the cytoplasmic signaling domain of T-cell receptors [254]. When chimeric antigen receptors bind to antigen-expressing target cells such as malignant tumors and CD19^+^ B cells, they induce cytolysis of the target cells and the differentiation of CART cells into long-lived memory CART cells [255,256,257]. Autoimmune B cells are generally activated and differentiate into PC, which divide into long-lived (LL) PC and short-lived (SL) PC [258,259]. LLPCs continue to live for years. However, SLPCs are usually only viable for a few days and are recruited by memory B cells. CD19 is expressed from the pro-B cell stage and is downregulated in SLPCs and LLPCs. However, a subset of PC maintains the expression of CD19, while CD20 is initially expressed at the pre-B cell stage and is downregulated on PC [259]. Because of this antigen expression pattern, anti-CD20 antibodies (such as rituximab) indirectly target SLPCs by eliminating progenitor memory B cells, but not LLPCs. CD19-targeted agents (e.g., CART cells) target pro-B cells and decrease both SLPC and LLPC. Since B cell maturation antigen (BCMA) is expressed on SLPC and LLPC and is essential for their survival, agents that target BCMA deplete PC. CD19-CART and BCMA-CART are currently approved for use clinically in the treatment of malignancies. In target B cells, loss or reduction of antigen expression has been observed with these CART cell therapies [260]. CART cells that target CD19/BCMA have also been developed [259]. It has not yet been evaluated in animal models of SS. However, a single-arm, open-label study (NCT05085431) evaluating the safety and efficacy of CD19/BCMA-CART cells in treatment-resistant SS is underway.

## 9. Conclusions

MSG is a valuable source of tissue for histopathological and genetic analyses of SS. Cytokines and chemokines produced by differentiated CD4^+^ T cells promote inflammation and damage salivary gland tissues, while CD8^+^ CTLs or CD4^+^ CTLs that recognize autoantigens impair exocrine functions with more direct cytotoxic effects. The SS-like disease that appears through the treatment of malignancies with ICIs indicates the important role of PD-1 on CTLs in the pathogenesis of pSS. Clinically, efforts should be made to reduce the side effects of exocrine stimulants that promote residual secretory function. As a localized therapy, salivary gland duct cleansing procedure for the major salivary glands, which is less invasive than endoscopy, represent an alternative approach. Studies with monoclonal antibodies against CD20, BAFF, BAFF receptors, CD22, CD40, and ICOSL are underway to block the signals involved in B cell proliferation and differentiation. Anti-BAFF receptor antibodies have been shown to improve systemic symptom indices, and targeted therapy is also important for identifying primary targets among the diverse cell subsets that are elevated in SS. Further research is needed to apply therapies that are already in the clinical stage for other autoimmune diseases, such as S1P receptor modulators, MSCs, and CART cells, to pSS.

## Figures and Tables

**Figure 1 ijms-24-11179-f001:**
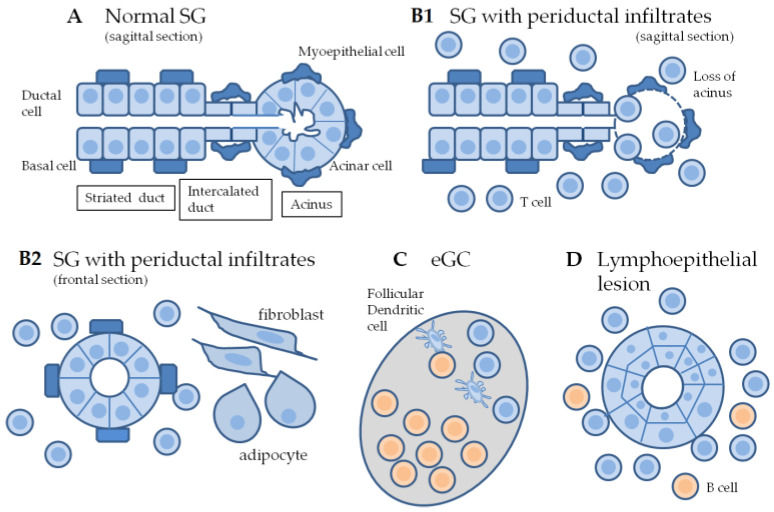
Changes in salivary gland histology with SS. The salivary gland acinus is composed of acinar cells and surrounding myoepithelial cells. Excretory ducts are further divided into intercalated ducts, striated ducts, and terminal excretory ducts (**A**). In the intercalated ducts, myoepithelial cells surround the ductal cells that form the lumen. The histological changes in the salivary glands in SS are dense periductal cellular infiltration and the loss of acinar cells (**B1**,**B2**). eGCs are formed in the salivary glands as a site for autoantibody production (**C**). Lymphoepithelial lesions consist of the hyperplasia of basal cells of the striated ducts and infiltrating lymphocytes surrounded by B cells (**D**). SG, salivary gland; eGC, ectopic germinal center.

**Figure 2 ijms-24-11179-f002:**
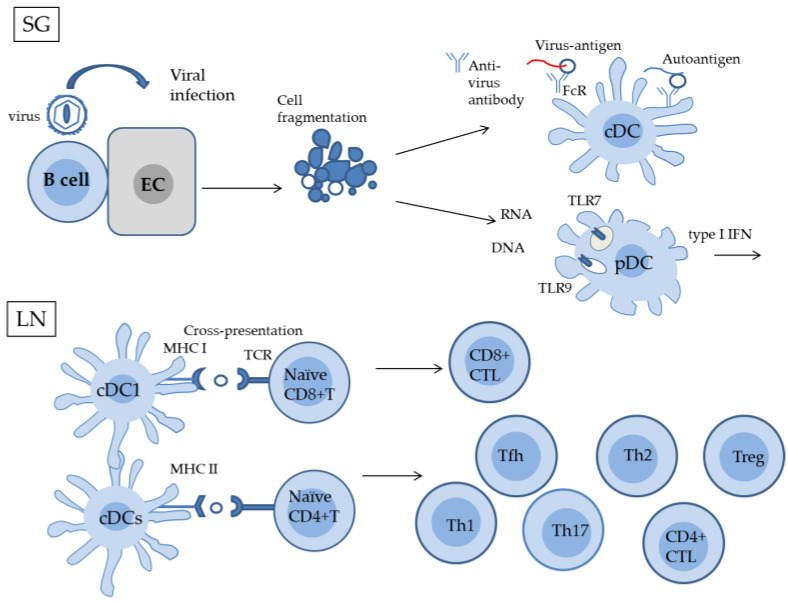
Proposed model of antigen uptake in salivary glands and autoreactive T cell differentiation in lymph nodes. Damage to salivary gland epithelial cells by viral infection releases autoantigens extracellularly, leading to their exposition to the immune system. Among these autoantigens, anti-viral antibodies react to a soluble autoantigen that has a common epitope with a specific viral antigen. Immune complexes formed by this reaction bind to the FcR of cDC and are taken up intracellularly. cDC may cross-present autoantigens in an MHC I-restricted manner to naive CD8^+^ T cells, resulting in their differentiation to CD8^+^ effector T cells (CD8^+^ CTL). On the other hand, MHC II-restricted antigen presentation from cDC causes naive CD4^+^ T cells to differentiate into Th1 cells, Th2 cells, Th17 cells, Treg, and Tfh cells. pDC recognizes self-nucleic acids as well as viral RNA and DNA through TRLs and produce type I IFNs. SG, salivary gland, EC, epithelial cell, TLR, Toll-like receptor; cDC, conventional DC; pDC, plasmacytoid DC; LN, lymph node; TCR, T cell receptor.

**Figure 3 ijms-24-11179-f003:**
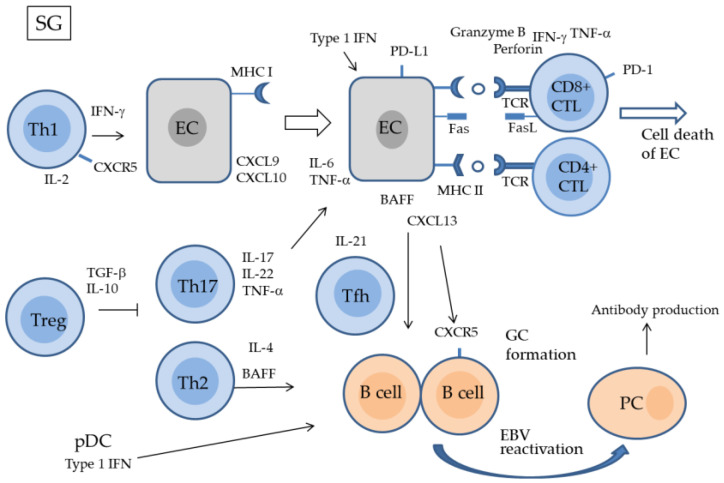
Epithelial cell damage by CD8^+^ T cells, CD4^+^ T cells, and cytokines in salivary glands. CD8^+^ CTLs recognize autoantigens presented by MHC I on salivary gland epithelial cells and destroy these cells by IFN-γ, TNF-α, and granzyme/perforin. The binding of FasL on CTLs to FAS or the binding of TNF-α to TNF receptors induces cell death. Since salivary gland cells are induced to express MHC II by Th1 cell-derived IFN-γ, they may become the target of CD4^+^ CTLs. Th1 produces IL-2, IFN-γ, and macrophage colony stimulating factor (M-CSF); Th2 produce IL-4, IL-5, BAFF, and mast cell growth factor (MCGF); Th17 cells produces IL-17 and IL-22; Tfh secretes IL-21 and contributes to B cell maturation. The expression of lytic cycle proteins due to EBV reactivation contributes to B cell immune evasion and proliferation. PD-1 on CTLs binds to PD-L1 on salivary gland cells and acts as a brake against T-cell-induced damage. SG, salivary gland; EC, epithelial cell; TCR, T cell receptor; pDC, plasmacytoid DC; PC, plasma cell; GC, germinal center.

**Figure 4 ijms-24-11179-f004:**
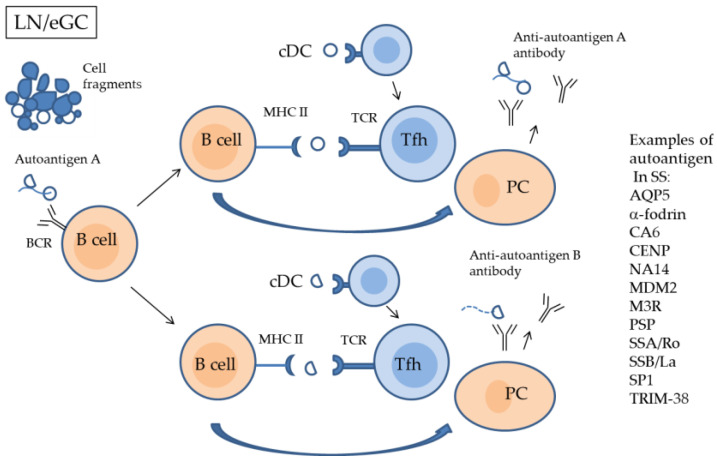
Proposed model of epitope spreading in lymph nodes and salivary glands eGCs. As described in Figure 2, autoantibodies against an autoantigen that shares a common epitope with a specific viral antigen are produced. B cells incorporate autoantigen A through BCR and present the second epitope of autoantigen A to Tfh. If there is a Tfh cell that recognizes this epitope, B cells differentiate into plasma cells and produce an antibody against autoantigen B with the second epitope. This epitope spreading process targeting additional autoepitopes will lead to the development of autoimmune diseases. The autoantigens reported to date in SS patients are shown on the right side of the figure. LN, lymph node; eGC, ectopic germinal center; PC, plasma cell; BCR, B cell receptor; TCR, T cell receptor; AQP5, aquaporin-5; CA6, carbonic anhydrase 6; CENP, centromere protein; NA14, nuclear autoantigen 14 kDa; MDM2, mouse double minute 2; M3R, muscarinic acetylcholine receptor 3; PSP, parotid secretory protein; SP-1, salivary protein-1; SSA/Ro, Sjögren’s syndrome-related antigen A/Ro; SSB/La, Sjögren’s syndrome-related antigen B/La; TRIM38, tripartite motif-containing protein 38.

**Figure 5 ijms-24-11179-f005:**
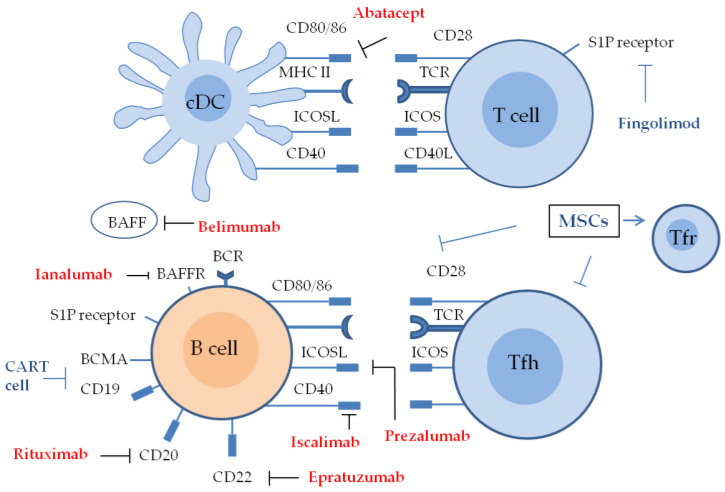
Biologic therapy for molecular targets in SS. Anti-B-cell agents include rituximab for CD20, epratuzumab for CD22, belimumab for BAFF, and ianalumab for BAFF receptors. Blockers of co-stimulation molecules include abatacept for CD80/86/CTLA4, iscalimab for CD40, and prezalumab for ICOSL. S1P/S1P1 modulators inhibit lymphocyte migration, because the release of lymphocytes from peripheral lymphoid organs into the circulation is mediated by the S1P receptor. MSCs decrease inflammatory responses by Th-17, and Tfh cells, and increase those of Tregs and Th2 cells. CART therapy targeting CD19 in B cells or BCMA in plasma cells. MSC, mesenchymal stem cell; S1P receptor, sphingosine-1-phosphate receptor; BAFFR, BAFF receptor; BCR, B cell receptor; TCR, T cell receptor; CART cell, chimeric antigen receptor T cell.

## Data Availability

Not applicable.

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
