# Peer review of "Outline of Salivary Gland Pathogenesis of Sjögren’s Syndrome and Current Therapeutic Approaches"

_ijms, 2023, doi:10.3390/ijms241311179_

Round 1

Reviewer 1 Report

In this manuscript the authors provide an interesting, well-structured, and comprehensive review of the pathogenesis of Sjögren syndrome (SS), as well as of the current and potential future therapeutic approaches. The thorough analysis of the pathophysiologic mechanisms implicated in the development of SS in the first part allows the reader to better comprehend all the aspects of therapeutic targeting that follow. There are though some points to be addressed for the sake of accuracy and some others to be included in order to achieve a more complete review, especially regarding the part referring to therapeutic approaches.

Comment 1

Page 1, line 39: “Blood tests detect an increase in circulating B-cell counts”

The authors should consider rephrasing this statement since research has demonstrated a peripheral B-cell subsets disturbance (e.g. Reduced number of CD27+ memory B cells, increased number of CD27- naïve B cells).

[Relative reviews: 1. B cell dysregulation in primary Sjögren’s syndrome: A review.

Hazim Mahmoud Ibrahem, doi: 10.1016/j.jdsr.2019.09.006, 2. The Multiple Roles of B Cells in the Pathogenesis of Sjögren’s Syndrome. Wenhan Du et al., https://doi.org/10.3389/fimmu.2021.684999]

Comment 2

Page 1, lines 41-42: “Among cases of pSS, 10-15% progress to the lymphoma stage”.

According to published meta-analyses of cohort studies, the percentage of lymphoma development seems to be lower (5-10%).

[1. The risk of lymphoma development in autoimmune diseases: a meta-analysis. Elias Zintzaras et al., DOI: 10.1001/archinte.165.20.2337,  2. Rate, risk factors and causes of mortality in patients with Sjögren's syndrome: a systematic review and meta-analysis of cohort studies. Singh Abha G. et al., doi: 10.1093/rheumatology/kev354]

Comment 3

Page 1, line 42: “develop MALT lymphoma”

This should be rephrased to “develop predominantly MALT lymphoma” given that patients with SS also develop other types of lymphoma.

[Relative review: Predisposing Factors, Clinical Picture, and Outcome of B-Cell Non-Hodgkin’s Lymphoma in Sjögren’s Syndrome. Ioanna E. Stergiou et al., https://doi.org/10.3390/immuno2040037]

Comment 4

Page 2, line 75: “(Fig.1)”, and Figure 1

It might be better for the ease of the reader to subdivide Figure 1 in parts (A, B, C, etc.) and refer to each of the described sections in the text.

Comment 5

Page 3, lines 114-115: “destruction of salivary gland epithelial cells by infiltrating T cells”

The authors should consider rephrasing. Though T cells are a major player in the destruction of salivary gland epithelial cells (as the authors nicely analyze in a following section of the manuscript), this destruction is not merely mediated by T cells. [e.g., an unknown trigger (viral infection or tissue damage) might drive epithelial cell apoptosis and senescence, loss of epithelial progenitor cells, the lymphocytic infiltration (both T and B cell) itself].

More relevant reference to be cited: reference no [8].

Comment 6

Page 5, lines 194-198.

This model of Jog et al. has been proposed for SLE. The theory of the common epitope is not confirmed for SS. So the authors should consider rephrasing. [Similar to comment 8]

Comment 7

In the manuscript section “3.5. Role of B cells and formation of eGCs” the authors should consider including a part on the clonality of B cells infiltrating the salivary gland lesions of patients with SS, since it has been demonstrated that these B cells express a certain rather limited BCR repertoire, while as the severity of the lesion progresses oligoclonal or even monoclonal B cell populations arise.

[Relative references: 1. Jordan, R.C. et al. High prevalence of B-cell monoclonality in labial gland biopsies of Japanese Sjögren's syndrome patients. Int J Hematol 1996, 64, 47-52, doi:10.1016/0925-5710(96)00462-8, 2. Miklos, J.A. et al. Salivary gland mucosa-associated lymphoid tissue lymphoma immunoglobulin VH genes show frequent use of V1-69 with distinctive CDR3 features. Blood 2000, 95, 3878-3884, doi:10.1182/blood.V95.12.3878, 3.  Visser, A. et al. Analysis of B-Cells Located in Striated Ducts of Salivary Glands of Patients With Sjögren's Syndrome. Frontiers in Immunology 2020, 11, doi:10.3389/fimmu.2020.01486.

Comment 8

Page 9, lines 391-393: “This 391 autoimmune response is not limited to the epitope that cross-reacts with the viral 392 protein with structural similarity (e.g., EBNA-1 with Sm B in SLE)” Both the references used refer to the case of SLE.

Do we have any evidence of such structural similarity in SS? If not, the above could only be assumed.  

Comment 9

Page 11, lines 488-491.

The following reference regarding the clinically significant difference in ESSDAI, the efficacy of an investigational drug, etc. should be added:

Seror R et al. Defining disease activity states and clinically meaningful improvement in primary Sjogren's syndrome with EULAR primary Sjogren's syndrome disease activity (ESSDAI) and patient-reported indexes (ESSPRI). DOI: 10.1136/annrheumdis-2014-206008

Comment 10

Manuscript section “7. Biological therapy for molecular targets”

The authors should consider dividing this section in two subsections: 7.1. Drugs targeting the BCR signaling pathway and 7.2. Monoclonal antibodies

Comment 11

Page 13, lines 620-622

This sentence should be in the previous paragraph, where the authors describe these drugs.

Comment 12

Figure 5

It would be easier for the reader if different colors were used to discriminate the treatments previously or currently tested and those that belong to the future prospectives.

Comment 13

The authors have not included some of the previously or currently tested drugs for SS. An indicative list of such drugs follows.

Amongst the ongoing clinical trials, some have already published preliminary results.

Therefore, it is should be considered to include them in the relevant parts of the manuscript sections 7 and 8

Anifrolumab: Anti-IFNAR mAb [https://clinicaltrials.gov/ct2/show/NCT05383677]

Baracitinib: JAK1/2 inhibitor [http://dx.doi.org/10.1136/annrheumdis-2021-222053]

Branebrutinib : BTK inhibitor [https://clinicaltrials.gov/ct2/show/NCT04186871]

Dazodalibep: CD40L blockade [http://dx.doi.org/10.1136/annrheumdis-2023-eular.234]

Efgartigimod: Anti-FcRn mAb [http://dx.doi.org/10.1136/annrheumdis-2023-eular.3188]

Iguratimod: NF-Κb blockade [DOI: 10.1080/03009742.2020.1809701]

Nipocalimab: Anti-FcRn mAb [https://clinicaltrials.gov/ct2/show/NCT04968912]

Seletalisib: PI3Kδ inhibitor [doi: 10.1093/rheumatology/keaa410]

Tofacitinib: JAK inhibitor [https://clinicaltrials.gov/ct2/show/NCT04496960]

Comment 14

In the Conclusion it would be preferable to refer to the exocrine stimulants and localized therapy first and afterwards in the biological therapy approaches, in order to follow the structure of the main text.

General Comment

Minor editing issues should be addressed

e.g.

- Page 2, line 67:myoepitelila” myoepithelial

- Page 4, line 189: “HMC II” MHC II

- Page 9, line 375: “focal score” focus score

- Page 14, line 625: “reported” → reported

- Page 16, line 763: “Matiette” → Mariette

Author Response

Referee 1

Thank you for careful reading and helpful suggestions.

Comment 1

Page 1, line 39: “Blood tests detect an increase in circulating B-cell counts”

The authors should consider rephrasing this statement since research has demonstrated a peripheral B-cell subsets disturbance (e.g. Reduced number of CD27+ memory B cells, increased number of CD27- naïve B cells).

[Relative reviews: 1. B cell dysregulation in primary Sjögren’s syndrome: A review.

Hazim Mahmoud Ibrahem, doi: 10.1016/j.jdsr.2019.09.006, 2. The Multiple Roles of B Cells in the Pathogenesis of Sjögren’s Syndrome. Wenhan Du et al., https://doi.org/10.3389/fimmu.2021.684999]

Response: Following this suggestion, the text was changed to simply state “Blood test detect peripheral B-cell subsets disturbance” (page 1, line 41)

Comment 2

Page 1, lines 41-42: “Among cases of pSS, 10-15% progress to the lymphoma stage”.

According to published meta-analyses of cohort studies, the percentage of lymphoma development seems to be lower (5-10%).

[1. The risk of lymphoma development in autoimmune diseases: a meta-analysis. Elias Zintzaras et al., DOI: 10.1001/archinte.165.20.2337,  2. Rate, risk factors and causes of mortality in patients with Sjögren's syndrome: a systematic review and meta-analysis of cohort studies. Singh Abha G. et al., doi: 10.1093/rheumatology/kev354]

Response: 10-15% was changed to 5-10%. This percentage is described in ref. 8, which is used as a reference in the citation (page 1, line 43).

Comment 3

Page 1, line 42: “develop MALT lymphoma”

This should be rephrased to “develop predominantly MALT lymphoma” given that patients with SS also develop other types of lymphoma.

[Relative review: Predisposing Factors, Clinical Picture, and Outcome of B-Cell Non-Hodgkin’s Lymphoma in Sjögren’s Syndrome. Ioanna E. Stergiou et al., https://doi.org/10.3390/immuno2040037]

Response : This sentence was changed to « -- and develop B-cell non-Hodgkin lymphomas, with MALT lymphoma predominantly” with the addition of reference (ref. 9) by Stergou et al. (page 2, lines 44-45)

Comment 4

Page 2, line 75: “(Fig.1)”, and Figure 1

It might be better for the ease of the reader to subdivide Figure 1 in parts (A, B, C, etc.) and refer to each of the described sections in the text.

Response: Following the suggestion, markers (A, B, C, D) were added to Figure 1, the legend, and the text (page 2, lines 69, 79, 85, 96).

Comment 5

Page 3, lines 114-115: “destruction of salivary gland epithelial cells by infiltrating T cells”

The authors should consider rephrasing. Though T cells are a major player in the destruction of salivary gland epithelial cells (as the authors nicely analyze in a following section of the manuscript), this destruction is not merely mediated by T cells. [e.g., an unknown trigger (viral infection or tissue damage) might drive epithelial cell apoptosis and senescence, loss of epithelial progenitor cells, the lymphocytic infiltration (both T and B cell) itself].

More relevant reference to be cited: reference no [8].

Response: “destruction of salivary gland epithelial cells by infiltrating T cells” has been changed to “destruction of salivary gland epithelial cells (page 3, line 117).

Comment 6

Page 5, lines 194-198.

This model of Jog et al. has been proposed for SLE. The theory of the common epitope is not confirmed for SS. So the authors should consider rephrasing. [Similar to comment 8]

Response: In the present revision, we described that Jog et al. proposed a model in a paper that overviewed EBV and autoimmune reactions in SLE. We then stated that the model may be applicable to SS, since anti-Ro and anti-La antibodies detected in SS patients react with EBV proteins (page 5, lines 200-205).

Comment 7

In the manuscript section “3.5. Role of B cells and formation of eGCs” the authors should consider including a part on the clonality of B cells infiltrating the salivary gland lesions of patients with SS, since it has been demonstrated that these B cells express a certain rather limited BCR repertoire, while as the severity of the lesion progresses oligoclonal or even monoclonal B cell populations arise.

[Relative references: 1. Jordan, R.C. et al. High prevalence of B-cell monoclonality in labial gland biopsies of Japanese Sjögren's syndrome patients. Int J Hematol 1996, 64, 47-52, doi:10.1016/0925-5710(96)00462-8, 2. Miklos, J.A. et al. Salivary gland mucosa-associated lymphoid tissue lymphoma immunoglobulin VH genes show frequent use of V1-69 with distinctive CDR3 features. Blood 2000, 95, 3878-3884, doi:10.1182/blood.V95.12.3878, 3.  Visser, A. et al. Analysis of B-Cells Located in Striated Ducts of Salivary Glands of Patients With Sjögren's Syndrome. Frontiers in Immunology 2020, 11, doi:10.3389/fimmu.2020.01486.

Response: In the revised version, we attempted to explain the presence of multiple salivary gland infiltrating B cell subsets and the expansion of a specific subset that develops lymphoma based on the description of Jordan et al, Visser et al, and Vernstappen et al (page 9, lines 399-406).

Comment 8

Page 9, lines 391-393: “This 391 autoimmune response is not limited to the epitope that cross-reacts with the viral 392 protein with structural similarity (e.g., EBNA-1 with Sm B in SLE)” Both the references used refer to the case of SLE.

Do we have any evidence of such structural similarity in SS? If not, the above could only be assumed.  

Response: Structural similarity is explained in comment 6. The portion “(e.g., EBNA-1 with Sm B in SLE) was deleted (Page 9, line 392).

Comment 9

Page 11, lines 488-491.

The following reference regarding the clinically significant difference in ESSDAI, the efficacy of an investigational drug, etc. should be added:

Seror R et al. Defining disease activity states and clinically meaningful improvement in primary Sjogren's syndrome with EULAR primary Sjogren's syndrome disease activity (ESSDAI) and patient-reported indexes (ESSPRI). DOI: 10.1136/annrheumdis-2014-206008

Response: Following the referee's suggestion, we have added the results reported by Seror et al. to the text (page 11, lines 483-487).

Comment 10

Manuscript section “7. Biological therapy for molecular targets”

The authors should consider dividing this section in two subsections: 7.1. Drugs targeting the BCR signaling pathway and 7.2. Monoclonal antibodies

Response: Since there are accumulated data on treatment with monoclonal antibodies, we addressed that topic. In the first part of section 7, we added brief comments on several new drugs proposed by the referee in comment 13 (page 14, lines 630-635, references 201 and 202).

Comment 11

Page 13, lines 620-622

This sentence should be in the previous paragraph, where the authors describe these drugs.

Response: As noted by the referee, this sentence should be included in the previous paragraph. This has been corrected (page 14, line 627).

Comment 12

Figure 5

It would be easier for the reader if different colors were used to discriminate the treatments previously or currently tested and those that belong to the future prospectives.

Response: In Figure 5, the color of the letters in the drug name was changed, with monoclonal antibodies in red and others in blue.

Comment 13

The authors have not included some of the previously or currently tested drugs for SS. An indicative list of such drugs follows.

Amongst the ongoing clinical trials, some have already published preliminary results.

Therefore, it is should be considered to include them in the relevant parts of the manuscript sections 7 and 8

Anifrolumab: Anti-IFNAR mAb [https://clinicaltrials.gov/ct2/show/NCT05383677]

Baracitinib: JAK1/2 inhibitor [http://dx.doi.org/10.1136/annrheumdis-2021-222053]

Branebrutinib : BTK inhibitor [https://clinicaltrials.gov/ct2/show/NCT04186871]

Dazodalibep: CD40L blockade [http://dx.doi.org/10.1136/annrheumdis-2023-eular.234]

Efgartigimod: Anti-FcRn mAb [http://dx.doi.org/10.1136/annrheumdis-2023-eular.3188]

Iguratimod: NF-Κb blockade [DOI: 10.1080/03009742.2020.1809701]

Nipocalimab: Anti-FcRn mAb [https://clinicaltrials.gov/ct2/show/NCT04968912]

Seletalisib: PI3Kδ inhibitor [doi: 10.1093/rheumatology/keaa410]

Tofacitinib: JAK inhibitor [https://clinicaltrials.gov/ct2/show/NCT04496960]

Response: Thank you for suggesting new candidates for SS treatment. The following four agents are in clinical trials and have not yielded results; therefore, they are not included in this review.

Anifrolumab: Anti-IFNAR mAb [https://clinicaltrials.gov/ct2/show/NCT05383677]

Branebrutinib : BTK inhibitor [https://clinicaltrials.gov/ct2/show/NCT04186871]

Tofacitinib: JAK inhibitor [https://clinicaltrials.gov/ct2/show/NCT04496960]

Nipocalimab: Anti-FcRn mAb [https://clinicaltrials.gov/ct2/show/NCT04968912]

Efgartigimod: For this CD40L blockade, there is a conference report describing clinical results, but the date and venue are unknown and are not included in this review.

Iguratimod: Since Iguratimod: is described in the original article, this report "Iguratimod: NF-Κb blockade [DOI: 10.1080/03009742.2020.1809701]" is not added to this review.

Baracitinib: JAK1/2 inhibitor [http://dx.doi.org/10.1136/annrheumdis-2021-222053]

Seletalisib: PI3Kδ inhibitor [doi: 10.1093/rheumatology/keaa410]

As noted in comment 10, these two papers related to JAK1/2 and PI3Kd have been added to the text and references (page 14, lines 630-635; references 201 and 202).

Comment 14

In the Conclusion it would be preferable to refer to the exocrine stimulants and localized therapy first and afterwards in the biological therapy approaches, in order to follow the structure of the main text.

Response: This has been rectified (page 19, lines 870-873).

General Comment

Minor editing issues should be addressed

e.g.

- Page 2, line 67: “myoepitelila” → myoepithelial

- Page 4, line 189: “HMC II” → MHC II

- Page 9, line 375: “focal score” → focus score

- Page 14, line 625: “reported” → reported

- Page 16, line 763: “Matiette” → Mariette

Response: Thank you for pointing this out. These errors have been corrected (page 2, line 71; page 4, line 190; page 9, line 374; page 14, line 638; page 16, line 771).

Reviewer 2 Report

This is not an 'outline' but a comprehensive review of SS and novel therapeutic approaches for SS. I suggest adjusting the title to reflect the scope of the manuscript.

A couple of additional suggestions: 

Streamline the introduction, especially section 2.  

Add info on potential new therapies that could target GCs in SS focusing on Tfh cells and GC -promoting cytokines

Quality of English language is OK, minor editing would suffice.

Author Response

Referee 2

Comments and Suggestions for Authors

This is not an 'outline' but a comprehensive review of SS and novel therapeutic approaches for SS. I suggest adjusting the title to reflect the scope of the manuscript.

Response: Thank you for your helpful suggestions. As pointed out by the referee, this review may cover a comprehensive area of salivary gland pathology in SS. However, there are many areas that are not explained. For example, the diverse B-cell subsets that contribute to the pathogenesis and the novel therapeutic agents currently in NCT clinical trials are not included in this review. Mechanistic explanations are also limited to those relevant to recent therapeutic approaches. Therefore, we wish to retain the term "overview" in this article title.

A couple of additional suggestions: 

Streamline the introduction, especially section 2.  

Response: To improve the introduction, the sentence "Basic knowledge of mechanisms and therapeutic targets is essential to better understand and utilize the expanding range of therapeutic agents." was added. (page 2, lines 58-60).

Add info on potential new therapies that could target GCs in SS focusing on Tfh cells and GC -promoting cytokines

 Response: As noted by the referee, many potential therapies are being tested in SS clinical trials (e.g., NCT). However, publication of the results of these studies is limited. This review focuses on studies that include clinical results from a sufficient number of patients. In addition, the majority of this review discusses therapeutics that target B cells. Examples include rituximab (anti-CD20 on B cells), epratuzumab (anti-CD22 on B cells), ianalumab (anti-BAFF receptor on B cells), iscalimab (anti-CD40 on B cells), prezalumab (anti-ICOSL on B cells), and inhibitors of JAK, BTK, SYK and PI3K. CD40-CD40L and ICOS-ICOSL interactions between B cells and Tfh cells are also inhibited by iscalimab and prezalumab, respectively (Figure 5 and Table 1). In this review, we have also included antibodies against BAFF, a B cell growth factor that is secreted by various cell types and promotes B cell survival during B cell immaturity and maturation.

Quality of English language is OK, minor editing would suffice.

Response: This manuscript has been checked by native speakers with scientific knowledge.

Round 2

Reviewer 1 Report

The authors efficiently answered all comments and made the relevant changes in their manuscript.